# Flow Disturbance Characterization of Highly Filled Thermoset Injection Molding Compounds behind an Obstacle and in a Spiral Flow Part

**DOI:** 10.3390/polym15142984

**Published:** 2023-07-08

**Authors:** Ngoc Tu Tran, Andreas Seefried, Michael Gehde, Jan Hirz, Dietmar Klaas

**Affiliations:** 1Department of Mechanical Engineering, Chemnitz University of Technology, 09126 Chemnitz, Germany; 2Baumgarten Automotive Technics GmbH, Carl-Benz-Straße 46, 57299 Burbach, Germany

**Keywords:** thermoset molding compounds, injection molding, plug flow, fountain flow, filler content, weld line, computer simulation, surface roughness, wall slip, pressure sensor, infrared temperature sensor

## Abstract

In the injection molding process, weld line regions occur when a molten polymer flow front is first separated and then rejoined. The position, length, and angle of weld lines are dependent on the gate location, injection speed, injection pressure, mold temperature, and, especially, the direction and degree of the polymer melt velocity in the mold-filling process. However, the wall surface velocity of the thermoset melt in the mold-filling process is not zero, which is not found for thermoplastic injection molding. The main reason leading to this difference is the slip phenomenon in the filling phase between the thermoset melt and the wall surface, which is directly affected by the filler content. In this study, commercial thermoset phenolic injection molding compounds with different amounts of filler were employed to investigate not only the mechanism of weld line formation and development behind an obstacle in the injection molding process but also the flow disturbance of the thermoset melt in the spiral flow part. In addition, the effect of the wall slip phenomenon on the flow disturbance characterization and the mechanism of weld lines of selected thermoset materials was carefully considered in this research. Furthermore, the generated material data sheet with the optimal developed reactive viscosity and curing kinetics model was imported into a commercial injection molding tool to predict the weld line formation as well as the mold-filling behavior of selected thermoset injection molding compounds, such as the flow length, cavity pressure profile, temperature distribution, and viscosity variation. The results obtained in this paper provide important academic knowledge about the flow disturbance behavior as well as its influence on the mechanism of weld line formation in the process of thermoset injection molding. Furthermore, the simulated results were compared with the experimental results, which helps provide an overview of the ability of computer simulation in the field of the reactive injection molding process.

## 1. Introduction

Thermoset materials are used in various applications in which high thermo-mechanical, chemical, and electrical properties are required [1,2]. These applications specifically include automotive, aerospace, and electronics applications [3,4]. Thermoset polymer parts could be processed by various methods, which are compression molding, transfer molding, injection compression molding, and injection molding. However, the injection molding method that is defined by a cycle and automated process for manufacturing identical plastic articles from a mold is the most widely used [5,6,7,8,9]. Small or very large parts can be manufactured by the injection molding method. In this process, the thermoset molding compounds are plasticized at temperatures between 90 and 100 °C and then injected into a hot mold with a temperature of 160 to 190 °C [9].

The weld line is a common problem that appears in the injection molding process. It is the line formed by two or more different melt fronts joining together with a sharp angle during the mold-filling stage. It decreases the strength of the final molded products and produces cosmetic defects [10]. Most previous publications related to weld line formation and weld line strength in injection-molded parts have been conducted on thermoplastic materials [11,12,13,14,15,16,17,18]. For example, there are three main factors that strongly influence the weld line strength of thermoplastics [19,20,21], including the high orientation of the macromolecules and fillers parallel to the weld line, the lack of diffusion of the macromolecules between two melt front surfaces, and stress concentrations because of notches on the surface next to the weld line. The influence of processing parameters on the thermoplastic weld line strength has been investigated [22,23]. It was found that the melt and mold temperature had a great influence on these properties. In addition, the reduction in the weld line strength of unreinforced, amorphous thermoplastics was investigated, analyzed, and calculated using a physical model of molecular diffusion [24]. The experimental results showed that a combination of a low holding pressure and high melt temperature should be selected to improve the strength of the weld line. Furthermore, the notch structure of polystyrene was studied by Tomari [25], and it was reported that different bonding strengths were dependent on the depth of the notches.

However, the previously stated knowledge of the mechanism of the weld line formation of thermoplastic materials could be only partially applicable for highly filled thermoset injection molding compounds [26]. Previously published articles have shown that the mold-filling behavior of highly filled thermoset injection molding compounds is completely different from thermoplastic materials. Specifically, the mold-filling characterization of these materials in the injection molding process is a plug flow [27,28,29,30,31,32] instead of fountain flow, which is found for thermoplastic materials. In the filling phase of the injection molding process, there is a strong slip phenomenon between the thermoset melt and the wall surface, which is not found for the injection molding of thermoplastic materials. In addition, the effect of the filler content; the processing condition, such as the mold temperature; the injection speed; and the surface roughness of both the cavity and the core on the polymer filling behavior in the thermoset injection molding process [32] was successfully investigated and analyzed using the mold printing method, as shown in Figure 1. The slip phenomenon between thermoset melt and the mold wall surface was studied and explained via an analysis of the visualizable movement of the thermoset melt dyed white on the surface of the injection-molded parts. All experimental results showed that the filler amount, injection speed, mold temperature, and surface roughness had a great influence on the wall slip phenomenon of phenolic thermoset injection molding compounds in the filling phase. A lower filler amount and injection speed and a higher mold temperature and surface roughness decreased the wall slip phenomenon of the thermoset melts. Because of this wall slip phenomenon, the velocity profile of the molten thermoset on the interface between the thermoset melt and the mold wall surface must be different from zero [32], which is completely different from the velocity profile of molten thermoplastic materials.(Figure 2) Consequently, the mechanism of weld line formation and the development of high-filled thermoset melt in the filling phase is different from the previously stated knowledge of the theory about the weld line formation of thermoplastic materials. However, this problem has not yet been carefully investigated or published in any scientific articles.

Nowadays, simulation software such as Moldex3D 2022 R3, Moldflow, and SigmaSoft is widely being employed to simulate all phases of the injection molding process [28,29]. Potential problems in the filling phase of the injection molding process, such as weld line positions, air traps, and sink marks, which usually appear on the molded parts, could be predicted. As a result, mistakes in the design process can be modified and the processing conditions, such as injection speed, mold temperature, injection pressure, and holding pressure, could be optimized, which helps us to save time as well as the manufacturing cost. In order to simulate all phases of the injection molding process, it is necessary to define a material data sheet that includes the heat capacity and thermal conductivity, viscosity data that consist of a viscosity model and fitted coefficients, and pressure–volume–temperature (PVT) data that must include a PVT model and fitted parameters. Especially for the reactive injection molding simulation is a curing kinetics model with fitted parameters required [28,29,30,31].
(1)$$\frac{\partial \rho }{\partial t}+\nabla \cdot \rho v=0$$
(2)$$\rho \frac{\partial v}{\partial t}=\rho g-\nabla p+\nabla \cdot \eta D-\rho v\cdot \nabla v$$
(3)$$c_{p}\left(\frac{\partial T}{\partial t}+v\cdot \nabla T\right)=\beta T\left(\frac{\partial p}{\partial t}+v\cdot \nabla p\right)+\eta {\dot{\gamma }}^{2}+k\cdot \nabla^{2}T$$

Fluid dynamic equations, including the conservation of mass, momentum, and energy, as shown from Equations 1 to 3, respectively, are employed to write the simulation code for characterizing the mold-filling behavior of the polymer melt in the injection molding process [28,29]. In these equations η is viscosity, ρ is density, *c_p_* is heat capacity, *k* is thermal conductivity, T is temperature, v is the velocity vector, g is the total body force per unit mass, and β is the coefficient of the volume expansion. It can be seen from these equations that there are symbols which represent the material data sheet for the simulation process. Therefore, this will have a huge impact on the accuracy of the predicted results [33] if there is any change in the material data sheet. To improve the simulated results a fitting tool is required that can be used to import the measured material data for the injection molding simulation software.

The material data sheet for the thermoplastic injection molding simulation process is always available from material manufacturers or already added in the material data bank of the commercial injection molding simulation software [28,29,30,31]. In contrast, the material data sheet of almost currently commercial highly filled thermoset injection molding compounds is unavailable from material suppliers and seldom embedded in the material data bank of commercial simulation software because the rheological and thermal properties, such as viscosity and curing kinetics behavior, are difficult to measure [28,29]. For example, the viscosity of thermoset materials is not only dependent on the temperature and shear rate, like thermoplastics materials, but also dependent on the curing behavior. In addition, if the material data of thermoset materials could be successfully measured, the modeling of rheological and thermal data for the reactive injection molding simulation process will require extensive knowledge not only for creating reactive viscosity models but also in the field of the optimization algorithm. These existing problems are being solved step-by-step by the authors [28,29,30,31].

With profound knowledge in the field of rheological and thermal properties [29,34,35], the viscosity, curing kinetics, thermal conductivity, and heat capacity of thermoset injection molding compounds have been successfully studied and measured by the authors [19,20,21]. In addition, based on the measured rheological and thermal data, the numerical method was developed to generate a material data sheet for the thermoset simulation process. This innovation won the special prize at the Moldex3D Global Innovation Talent Award 2018. These fitted processing coefficients were integrated into a cure kinetics model, the Kamal model, and a reactive viscosity model, the Cross–Castro–Macosko model, which were used to simulate the reactive injection molding process [28,29].

Although a complete way to create thermoset material data from measured experimental data (thermal data and rheological data) for the reactive injection molding simulation process was successful studied by the authors [29], there are things which should be further studied. In the process of creating material data, the developed numerical method is based only on a cure kinetic model, namely the Kamal model, and a reactive viscosity model, namely the Cross–Castro–Macosko model, while there are still other cure kinetics and reactive viscosity models. In contrast, a comparison of the efficiency of using each cure kinetics model and reactive viscosity model to describe cure kinetics data and rheological data, respectively, has not yet been conducted. If this could be achieved, a thermoset material data sheet for the reactive simulation process could be created with the best cure kinetics and reactive viscosity model. In order to solve this problem, a complete fitting tool, namely Thermoset—TU—Fitting Tool [30,31], was successfully developed and written. In the writing process, the least-square estimation algorithm developed by Levenberg–Marquardt (LMA) [36,37] and embedded in Matlab program language was used. The Thermoset—TU—Fitting Tool was employed as a useful tool for transporting the experimental rheological and thermal data to any injection molding simulation software. With the Thermoset—TU—Fitting tool, an evaluation of developed reactive viscosity and cure kinetics models that are currently used for rheological and thermal simulation in the thermoset injection molding process was successfully carried out. The reactive viscosity models include the Castro–Macosko model, Cross–Castro–Macosko model, power-law Castro–Macosko model, and Herschel–Bulkley–WLF–Castro–Macosko model [30,31]. The cure kinetic models consist of the Kamal model, modified Kamal model, Deng Isayev model, and Grindling model [30,31].

By dint of using the developed Thermoset—TU—Fitting tool [30,31], it was found that all presented reactive viscosity and cure kinetics models could be used to describe the reactive viscosity and cure kinetics data well. In the case of curing kinetics models, the best curing model is still the previously used Kamal model. However, the reactive viscosity model, Herschel–Bulkley–WLF–Castro–Macosko model (Herschel–Bulkley model), describes and fits the the parabolic curve of the expemental viscosity property the best instead of the previously used Cross–Castro–Macosko model because the viscosity of selected thermoset materials at a low temperature is successfully simulated by the Herschel–Bulkley–WLF–Castro–Macosko model, which is not the case in other reactive viscosity models. The main reason that leads to the difference in the adaptation of reactive viscosity models in the characterization of viscosity is the yield stress phenomenon of high-filled plastics [31]. In the Herschel–Bulkley–WLF–Castro–Macosko model, there is a coefficient ($\tau_{y}=\tau_{y0}\cdot exp\left(\frac{T_{y}}{T}\right)$) that shows the influence of the yield stress phenomenon on the viscosity of high-filled thermoset injection molding compounds. Therefore, the generated Herschel–Bulkley–WLF–Castro–Macosko model describes and fits the experimental reactive viscosity data of high-filled thermoset materials the best. Consequently, the optimal reactive viscosity model and cure kinetics model were found and employed to generate the material data sheet of commercial high-filled thermoset materials for the reactive simulation process [30,31].

Based on the gained results and the existing problems, the present article focuses on two scientific key goals. On the one hand, the aim is to continuously understand and explain the physical filling behavior of reinforced thermoset injection molding compounds such as flow length, the influence of the wall slip and flow disturbance behavior on the mechanism of weld line formation, the pressure gradient, the temperature distribution, the viscosity characterization, and the degree of cure. On the other hand, the generated material data sheet with the optimal reactive viscosity and cure kinetics model will be employed to investigate the application of the commercial injection molding simulation software in the simulation of the mold-filling behavior of highly filled thermoset injection molding compounds.

## 2. Materials and Methods

### 2.1. Injection Molding Process

#### 2.1.1. Highly Filled Thermoset Injection Molding Compounds

Bakelite PF6680, Bakelite PF6506, and Bakelite PF1110 are three commercial thermoset phenolic injection molding compounds with different filler contents for the injection molding process, which were selected and ordered from a material supplier. The filler content is between 55 and 80 percent by weight, as shown in Table 1. GF means glass fiber and GB means glass ball.

#### 2.1.2. Studying Objects

The study objects are three different parts. The first part is the plate part with a hole (Figure 3). The dimensions of the plate part are 150 mm × 150 mm × 4 mm. The diameter of the hole as an obstacle is 8 mm. The plate part with the obstacle of 8 mm was used to investigate the influence of the filler content, the wall slip phenomenon, and the processing conditions on the mechanism of weld line formation and development in the mold-filling process. The second part is the spiral flow part (Figure 4) which was used to investigate the flow disturbance characterization such the flow length, pressure gradient, temperature, and viscosity behavior. The last part is a complex part from industry, which was employed to study both flow disturbance characterization and weld line positions.

#### 2.1.3. Experimental Procedure

A hydraulic Krauss Maffei injection molding machine KM 150–460B, with a screw diameter of 45 mm and a three-zone plasticizing cylinder, was employed for the injection molding process of the plate part with the obstacle and the spiral flow part.

Firstly, a simple standard two-plate mold was employed to study the weld lines formation mechanism of selected materials. In the experimental process, the mold painting method developed by the authors and published in the previous article continued to be employed [32]. A white mark was painted on a fixed location of the mold wall surface. The schematic position of the constant rectangular white mark painted on the wall surface of the mold is shown in Figure 1. The distance (a) of 20 mm from the location of the white mark to the boundary line between the cavity and the film gate was kept constant in all experimental processes. The injection molding experiments were conducted with three different phenolic injection molding compounds. The temperature profile in the injection chamber (cylinder temperature) is 100 °C–80 °C–60 °C. The mold temperature of 175 °C was kept constant under different injection speeds of 8 cm^3^/s, 16 cm^3^/s, and 32 cm^3^/s. In order to obtain more information for analyzing the mechanism of weld lines formation and development behind the obstacle, a series of incomplete molded parts with different percentages of cavity volume were made.

Secondly, to further analyze the flow disturbance behavior and the wall slip phenomenon of highly filled thermoset injection molding compounds, the spiral flow part with a flow length of 1385 mm was used for the subsequent experiments. With the spiral flow part, it is possible to study the influence of the chemical reaction on the viscosity, which also affects the flow length. In addition, the variation in the polymer temperature and pressure during the filling phase was analyzed using pressure and infrared temperature sensors that were mounted on the interface between the thermoset melt and the wall surface. Based on the temperature gradient, the generation of heat by chemical reactions and the heat transfer from the mold to the polymer melt was analyzed. In this step, only two phenolic injection molding compounds with the lowest and highest filler content—Bakelite PF6680 (55% filler) and Bakelite PF1110 (80% filler), respectively—were selected to conduct the spiral injection molding experiments. The temperature profile in the barrel is 100 °C–80 °C–60 °C. The injection speed profile in the screw is 6–10–12 cm^3^/s, which was kept constant. The mold temperatures are 160 °C, 175 °C, and 190 °C, respectively.

Finally, a complex industrial part from Baumgarten automotive technics GmbH, Carl-Benz-Straße 46, 57299 Burbach, Germany, was employed to study the flow length as well as the location of the weld lines. For commercial reasons, information about the material, the geometry of the complex industrial part, and the processing conditions will not be disclosed in this article. For information about the manufacturing process of the complex industrial part, please contact Baumgarten automotive technics GmbH by email: info@bat-duro.com

### 2.2. Simulation Process

#### 2.2.1. Generating Material Data Sheet for the Injection Molding Simulation Process

Since the material data sheets of Bakelite PF6680, Bakelite PF6506, and Bakelite PF1110 are not yet available from the material manufacturers and cannot be found in the data bank of any commercial injection molding simulation software, the Thermoset—TU—Fitting Tool developed by the authors [30,31] was employed to generate the material data sheet of these selected materials. In the generated material data sheet for the simulation process, the optimal reactive viscosity model, the Herschel–Bulkley–WLF–Castro–Macosko model with fitted coefficients, and the optimal cure kinetics model, the Kamal model with fitted coefficients, were employed to characterize the rheological and thermal properties, as shown in Figure 5 and Figure 6. To find out more information in the field of creating material data sheets for thermoset injection molding compounds, please refer to the international articles that were previously published by the authors [28,29,30,31].

The task in this working package is therefore to export the generated material data sheet of these selected materials as the input file for the next step, the simulation process. An input file was generated for each material which contains the value of heat capacity and the value of thermal conductivity at different temperatures. This file was cure kinetics data that include the Kamal model with fitted coefficients and the reactive viscosity model, the Herschel–Bulkley–WLF–Castro–Macosko model with fitted coefficients. All input files were imported in the commercial simulation software (Moldex3D) to predict the mold-filling behavior of the selected thermoset materials.

#### 2.2.2. Simulation of the Mold-Filling Behavior

Moldex3D 2022 R3 will be selected to simulate the reactive injection molding process since it implemented a high-performance finite volume method (HPFVM). This numerical method synergizes robustness and efficiency, in contrast to the finite element method [29,38]. The HPFVM in Moldex3D is called designer boundary layer mesh (BLM). This technique solves the transient flow field in three dimensions. It generates multiple layers of prismatic meshes inward from the surface mesh and then fills up the remaining space with a tetrahedral mesh. BLM can precisely capture the drastic changes in temperature and velocity near the cavity wall during the filling process. It can also help detect viscous heating and warpage problems in advance with accuracy. In addition, with Moldex3D, the wall slip boundary condition is considered during the simulation process.

The simulation subjects are firstly the plate part (Figure 3) and then the spiral part (Figure 4), which were also used in the injection molding process. The processing conditions are the same as the experimental process. In addition, to show the practical benefit of both the generated material data sheet and injection simulation model, the simulation process is supposed to test the complex industrial part from Baumgarten automotive technics GmbH.

## 3. Results and Discussion

### 3.1. Mechanism of Weld Line Formation and Development behind an Obstacle

It can be seen from Figure 7, Figure 8 and Figure 9 that the thermoset melt dyed a white color on the surface of the injected parts moves from the original painted position to near the melt front, a phenomenon also observed in previous publications [27,29,32]. The movement of the polymer dyed a white color is attributed to the wall slip between the phenolic polymer and the mold wall surface. Nevertheless, the intensities of the white stripes are different on the surface of the molded parts with different percentages of reinforced filler. Specifically, with the material PF6680 having the lowest filler content (55% filler), the white color still appears at the original painted position, as shown in Figure 7. However, for PF6506 with only 5% more filler, the density of the white color (Figure 8) at the original painted position is lower than and not as clear as PF6680. In contrast, in the material PF1110 with the highest filler content (80%), the white color does not appear at the original painted position (Figure 9). Based on the difference in the thermoset melt dyed white color at the original painted position, it can be summarized that PF1110 (GF35 + GB45) has the strongest slip phenomenon, followed by PF6506 (GF30 + GB30) and PF6680 (GF25 + GB30). In addition, the amount of filler is the main factor that has a great impact on the wall slip phenomenon at the interface between the phenolic polymer and the mold wall surface. The wall slip phenomenon is stronger if the amount of filler is higher.

With the same processing conditions, the joining mechanisms of the melt fronts of the selected highly filled thermoset injection molding compounds behind the obstacle in the filling phase are different. Beginning with 40% cavity volume, the melt fronts of PF6680 (Figure 7) and PF6506 (Figure 8) immediately reunite behind the obstacle, which is not the case with the mold-filling behavior of PF1110 (Figure 9). When the cavity volume increases by only 10%, the joined fronts of PF6680 (Figure 7, 50% cavity) and PF6506 (Figure 8, 50% cavity) slip in the flow direction. In contrast, the melt fronts of PF1110 are still separated and exhibit a translation motion in the flow direction. As a result, there is still a gap without polymer between two melt fronts (Figure 9, 50% cavity). The distance of the gap in the incomplete part molded from 50% cavity is at least as large as the diameter of the obstacle (8 mm).

When increasing the injection volume to 60%, 70%, 80%, and 90%, the reunited melt fronts of PF6680 and PF6506 continue to move slowly in the flow direction. The position of the weld line is found in the middle line of the complete molded part (100% Cavity), as shown in Figure 7 and Figure 8. However, this phenomenon was not observed in the melt front behavior of PF1110. When the cavity volume is 60% the melt fronts of PF1110 flow and slip not only in the flow direction but also perpendicular to the flow direction. Therefore, the gap without polymer starts narrowing. Nevertheless, the two melt fronts were still separated and began to join each other as the cavity volume increased (Figure 9 with 70% cavity and 80% cavity). The joining process of the melt fronts is not complete because there are still different air gaps without polymer on the surface of the molded parts, which means that there will be different small weld line regions on the surface of the complete plate molded part. Furthermore, the surface of this incomplete molded part is rough, and the molded parts are uncompacted. The melt fronts of PF1110 are completely joined with cavity volumes of more than 90%, which have an anisotropic motion.

### 3.2. Influence of Injection Speeds on the Mechanisim of Weld Line Formation and Development

Figure 10 and Figure 11 show that although there is change in the injection speed, the melt fronts of PF6680 und PF6506 are still immediately reunited behind the obstacle. The joined melt fronts slightly slip and move in the flow direction when the injection volume increases, as shown in Figure 12 and Figure 13. For all the investigated injection speeds, the weld line region is found in the middle line of the plate molded part. The density of the white color on the surface of the molded parts shows that the slip degree of the joined melt fronts is stronger when the injection speed increases.

In contrast to PF6680 und PF6506, the injection speed has a great impact on the weld- line formation mechanism of PF1110 (Figure 14). At the lowest injection speed of 8 cm^3^/s, the melt fronts behind the obstacle flow and slip straight in the flow direction. Therefore, the weld line is not yet formed, and there is a gap between the two melt fronts. As the injection speed increases, the gap between the two melt front surfaces slightly decreases. At the highest injection speed of 32 cm^3^/s, the melt fronts behind the obstacle start to reunite. These differences could be explained by analyzing the influence of the injection speed on the shear rate and slip phenomenon. The shear rate rises significantly with an increasing injection speed. Therefore, the polymer molecules between the different layers are separated and move more easily in a different direction, which together with the high slip velocity causes the melt fronts to move not only parallel to the flow direction but also in other directions. Consequently, at a higher injection rate, such as 32 cm^3^/s, the melt fronts of PF1110 merge immediately behind the obstacle. The joined melt fronts slip strongly in the flow direction (Figure 15). Moreover, as the injection speed increases, the zone with the compacted polymer that is located near the gate begins to appear. The surface of the molded part in this zone is smooth. In addition, it is noteworthy that at the highest injection speed (32 cm^3^/s), the joined melt fronts appear only on the compacted zone and slip to the end of the molded part, as shown in Figure 16.

From the experimental results presented in Section 3.1, it can be inferred that the weld line region is strongly dependent on the degree of wall slip that is directly affected by the filler content and processing conditions. For the thermoset injection molding materials with a filler content of less than 65%, such as PF6680 (GF25 + GB30) and PF6506 (GF30 + GB30), the influence of the injection speed on the mechanism of the weld line is not as great as that with a filler content of more than 65%, such as PF1110 (GF35 + GB45). It is noteworthy that the melt fronts of all the investigated thermoset materials behind the obstacle at a high injection velocity (for instance 32 cm^3^/s) immediately merge and continue to slip in the flow direction. Since increasing either the filler content or the injection speed would lead to a stronger slip of the joined melt fronts, the injection speed should be applied by taking into account both the amount of filler content of the thermoset injection molding compounds and the expected weld line regions of the manufacturer.

### 3.3. Mold-Filling Behavior in the Spiral Flow Part

#### 3.3.1. Flow Length

For all the investigated processing conditions, the spiral flow part with a maximum flow length of 1385 mm was molded from both materials, PF6680 and PF1110 (Figure 17). The mold-filling behavior with respect to the flow length of the incomplete molded parts is shown in Figure 18 and Figure 19. Based on the surface of spiral molded parts, it was found that the incomplete molded part of PF6680 is completely compacted (Figure 18), and there is not a zone with an uncompacted thermoset melt. However, for PF1110, an uncompacted zone with a rough surface (Figure 19) was observed again near the melt front, which also appeared on the surface of the incomplete plate molded parts, as previously shown in Figure 14, Figure 15 and Figure 16. Nevertheless, the length of the uncompacted zone on the surface of the spiral part is much shorter than was found on the surface of the plate part. This experimental result shows that besides the influence of processing conditions on the uncompacted zone of highly filled thermoset injection molding compounds, there is also the effect of the part geometry and the types of the gate. In this article, the film gate (Figure 3) was employed for the plate part, while the direct sprue gate (Figure 4) was applied for the spiral flow part.

It can be seen from Figure 19 that the uncompacted zone on the incomplete part molded from the lowest cavity volume (35% cavity) continues to appear at the end of the incomplete molded parts with higher cavity volumes (57% and 77% cavity) and disappears on the complete molded spiral flow part (Figure 17). This experimental result is explained by the plug flow behavior of PF1110 during the mold-filling process. The polymer region of PF1110 originated from the initial polymer portion which touched the mold surface which continuously flowed to the end of the cavity. Therefore, the velocity of polymer melts on the interface between the thermoset melt and mold wall surface is nonzero, which is usually the case for the mold-filling behavior of thermoplastic materials. Because of the wall slip in the filling process, a slip friction coefficient exists on the interface between the thermoset melt and the wall surface, as well as between different thermoset layers across the thickness of the cavity. The slip frictional coefficient has a great effect on the cavity pressure profile and melt temperature distribution results that are presented in the following subsection.

#### 3.3.2. Cavity Pressure Profile and Melt Temperature Distribution

Pressure and infrared sensors (Figure 4) were used in the experiments of the spiral flow part to study the influence of the flow disturbance that derives from the wall slip phenomenon. The typical experimental result of the cavity pressure profile is shown in Figure 20 and Figure 21. In the filling phase, the pressure drops per unit of length along the spiral flow path. The drop of pressure from one location to another creates a force that pushes the molten polymer to flow during the mold-filling process. Polymer always moves from a higher pressure to lower pressure, like water flowing from higher elevations to lower elevations. As a result, the maximum pressure is found in sensor P1 and the minimum pressure in sensor P6. However, there is a difference between PF6680 and PF1110 in the cavity pressure profile during mold-filling under various investigated mold temperatures (Figure 22 and Figure 23).

The required injection pressure to push the molten thermoset flow during the filling phase is proportional to the viscosity that is strongly dependent on the shear rate, temperature, and the degree of cure. The injection pressure of PF6680 (Figure 22) decreases when the mold temperature increases. This experimental result shows that in the mold-filling process, the viscosity of PF6680 is mainly dependent on the temperature and shear rate like any other thermoplastic materials. The degree of cure of PF6680 in the filling phase is very small, or the curing process has not even started yet. Therefore, as the temperature in the filling phase increases, the viscosity of PF6680 decreases, leading to a decrease in the required injection pressure. However, the cavity pressure profile of PF1110 in Figure 23 is different from that of PF6680. In the case of PF1110, during the first 6.3 s of the filling phase, the injection pressures at the mold temperatures of 175 °C and 190 °C do not differ significantly, as opposed to the cavity pressure profile of PF6680, which exhibits a stronger dependency on the mold temperature. Furthermore, the injection pressure at the mold temperature of 190 °C from 6.3 s to the end of the filling phase (9 s) is higher than the injection pressure at the mold temperature of 175 °C. At the end of the filling process, the maximum injection pressure is noticed at the highest mold temperature of 190 °C, followed by at the mold temperature of 175 °C. The minimum injection pressure is found at the lowest mold temperature of 160 °C.

The difference in the injection pressure profile between PF6680 and PF1110 could be explained based on mold-filling characterization. The mold-filling behavior of PF1110 is complete plug flow. Therefore, the polymer region of PF1110 (the uncompacted zone) which originated from the initial polymer portion which touched the mold surface continues to flow to the end of cavity, as shown in Figure 19. As a result, the residence time of the initial polymer portion in the mold is higher than the fresh molten polymer portion that is newly injected into the mold. The temperature of the initial polymer portion quickly increases during the flowing process because of the heat transfer from the hot mold to the molten polymer. Therefore, the melt temperature measured by the infrared temperature sensor 1 (T1) in the filling phase is always lower than the melt temperature measured by the infrared temperature sensor 2 (T2), as shown in Figure 24. In addition, it was found from Figure 25 that the melt temperature of PF1110 is higher than the melt temperature of PF6680 because the degree of the wall slip of PF1110 is greater than that of PF6680.

At the beginning of the filling phase, the residence time and temperature of the initial polymer portion are not high enough to start the curing process. Therefore, the viscosity of PF1110 at the beginning of the filling phase mainly depends on the temperature and shear rate, which is also the case with the mold-filling behavior of PF6680. After the 6.3 s period, the residence time of the initial portion in the mold is long enough for the temperature of the initial polymer portion to reach the temperature necessary to begin the curing kinetic process due to the continuous heat transfer process from the hot mold to the molten polymer. Therefore, the viscosity of PF1110 is now dependent mainly on the degree of cure, besides the temperature and shear rate. When the curing process begins, the viscosity of the thermoset melt significantly rises. The curing process in the filling phase is undesirable because it increases the viscosity too fast, rendering the initial polymer portion slightly solid, which could lead to problems such as flow hesitation or over-packing that results in flash. When the mold temperature (Figure 26) is higher than 175 °C, the melt temperature of PF 1110 in the filling phase rises drastically, enabling the curing process in the filling phase to start more easily. As a result, the viscosity of PF1110 begins rising slowly from 6.3 s to the rest of the injection molding time, which causes a non-uniform injection pressure at the end of the filling phase, as shown in Figure 23.

### 3.4. Validating Simulation Results and Adapting Simulation Model

The experimental results about weld line formation and development of highly filled thermoset injection molding compounds behind the obstacle are presented and analyzed in Section 3.1 and Section 3.2. Essentially, the wall slips phenomenon, the filler content, and process condition have a great impact on the joining process of the thermoset melt front the surfaces. For the thermoset injection molding materials with a filler content of less than 65%, such as PF6680 (GF25 + GB30) and PF6506 (GF30 + GB30), the melt fronts immediately reunite behind the obstacle. The weld line is formed in the middle of the plate part. The location of the experimental weld line is similar to the prediction made by the simulation tool, as shown in Figure 27a and Figure 28. In addition, the weld line positions of the complex industrial part predicted by the simulation tool are also in accordance with the experimental results that were produced by experts in the field of thermoset injection molding from Baumgarten automotive technics GmbH (Figure 29). Furthermore, Figure 27b shows that the experimental flow length of the spiral flow part under different mold temperatures is accurately predicted by the simulation tool. The simulated influence of the mold temperatures on the viscosity, which directly affects the injection pressure, also matches the experimental result (Figure 30). Most notably, both simulation and experimental results show that the higher the mold temperature, the lower the viscosity of PF6680 in the filling phase, leading to a decrease in the required injection pressure. The similarities between the simulated and experimental results suggest that the generated reactive viscosity data with the optimally developed viscosity model (Herschel–Bulkley–WLF–Castro–Macosko model) and curing kinetics model (Kamal model), which were previously presented in Figure 5 and Figure 6, are reasonable.

However, for the highly filled thermoset injection molding compounds (the filler content more than 65%), such as PF1110 (GF35 + GB45), the weld line formation mechanism is different from that of the lowly filled thermoset injection molding compounds, like PF6680 (GF25 + GB30) and PF6506 (GF30 + GB30). The melt fronts are not completely reunited when the cavity volume is still below 100%. There are always two zones on the surface of the incomplete molded part: the compacted zone located near the gate (Figure 15) and the uncompacted zone located next to the compacted zone which extends to the melt front. The weld line positions are dependent on the injection speed, which has a great impact on the wall slip. All these experimental results are not seen in the simulated results (Figure 31). In addition, the uncompacted zone of the incomplete molded parts that is strongly dependent on the injection speeds and the mold temperatures, has not yet been accurately simulated, as shown in Figure 31. Furthermore, the simulation tool fails to predict the non-uniform injection pressure due to the curing behavior of PF1110 that is found in the experimental injection molding process of the spiral flow part (Figure 23). Specifically, it can be seen from Figure 32 that the experimental injection pressure at the end of the filling process is proportional to the mold temperature, which is not the case in the simulation results. The main reason for these discrepancies is the influence of the wall slip phenomenon that generates friction between thermoset melt and the wall surface, which has a great effect on the mold-filling characterization. However, the slip frictional coefficient cannot yet be accurately estimated in most of today’s commercial injection molding software. These discrepancies are currently being studied and resolved by the authors.

## 4. Conclusions

In this study, the mold-filling behavior of highly filled thermoset injection molding compounds in the injection molding process, such as the weld line formation, reactive viscosity behavior, cavity pressure profile, and temperature distribution, is thoroughly investigated. It is found that this behavior is strongly dependent on the filler content, the processing conditions, and the wall slip phenomenon. The optimal injection speed and mold temperature applied in the injection molding process should be based on the overall analysis of the manufacturer’s expected weld line areas, the filler content reinforced for the thermoset material injection molding compounds, the geometry of the industrial parts, the type of injection gate, as well as the gate location. For the thermoset injection molding compounds with filler content of less than 65%, the effect of the wall slip phenomenon and processing conditions on the mechanism of the weld line formation is negligible. The generated viscosity with the optimally developed viscosity model, the Herschel–Bulkley–WLF–Castro–Macosko model, and the curing kinetics model, the Kamal model, was imported into the commercial injection molding simulation tool to simulate the form filling behavior of these materials. However, for the thermoset injection molding compounds with filler content of more than 65%, the wall slip phenomenon, mold temperature, and injection speed have a great impact on certain mold-filling characteristics (such as the formation and development of weld lines, compacted and uncompacted zones, pressure gradient, and the curing behavior in the filling process). This dependency has not yet been accurately simulated by the commercial simulation software, posing a new challenge to the fluid dynamic simulation community.

## Figures and Tables

**Figure 1 polymers-15-02984-f001:**
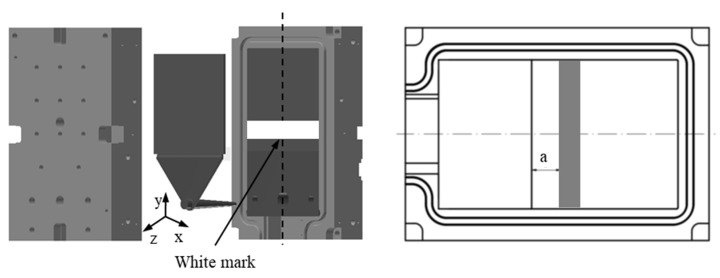
A simple standard two-plate mold with the location of painted white marks [32].

**Figure 2 polymers-15-02984-f002:**
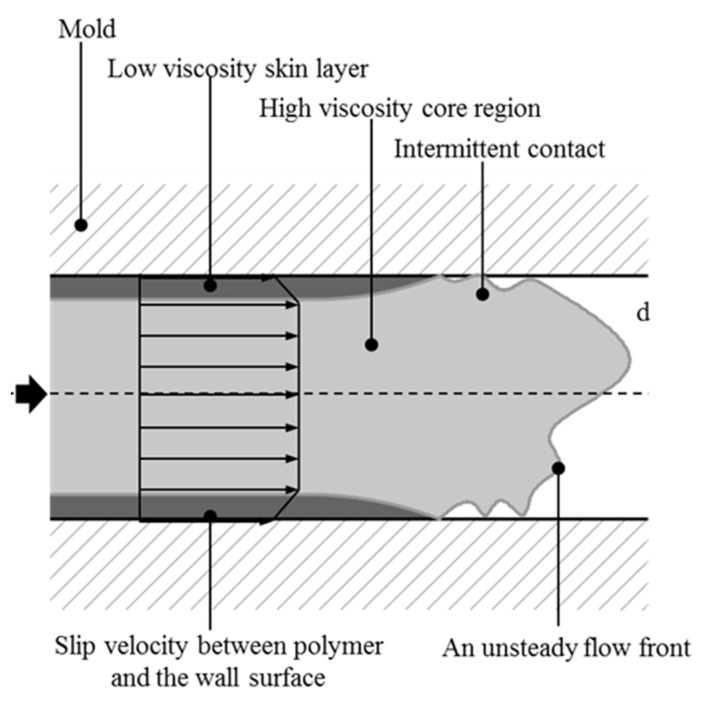
Velocity behavior of high-filled thermoset injection molding compounds in the mold-filling process [32].

**Figure 3 polymers-15-02984-f003:**
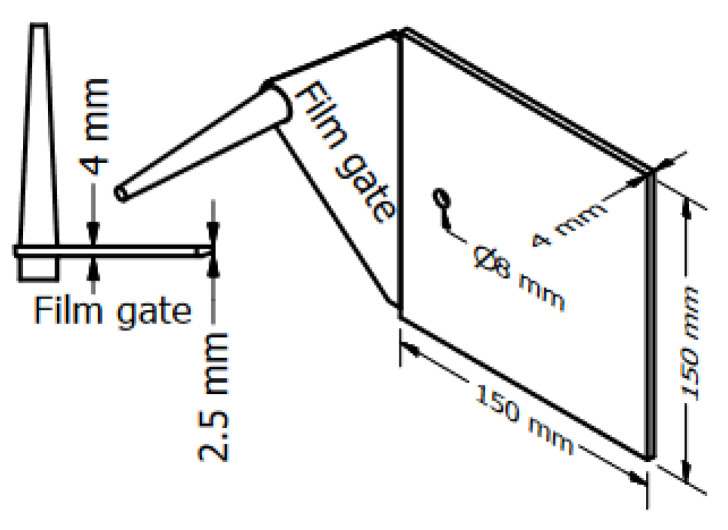
The plate part with a film gate and an obstacle.

**Figure 4 polymers-15-02984-f004:**
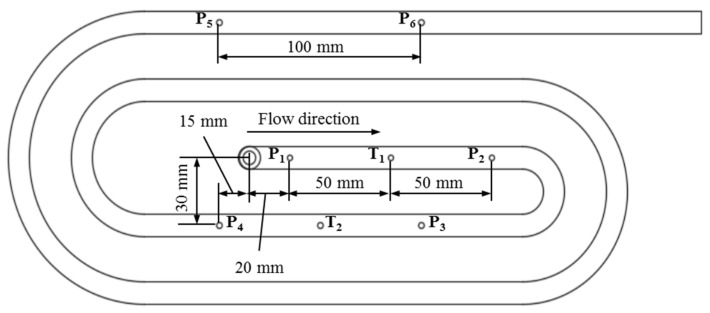
The spiral part; the location of six pressure sensors from P_1_ to P_6_ and two infrared temperature sensors T1 and T2.

**Figure 5 polymers-15-02984-f005:**
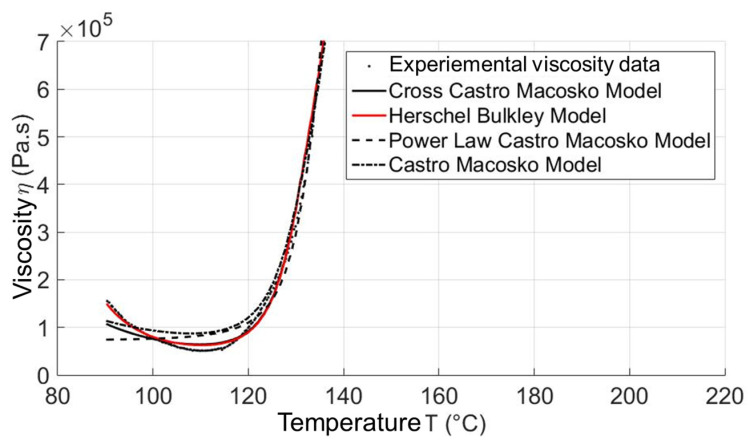
Modeling of reactive viscosity models for the injection molding simulation.

**Figure 6 polymers-15-02984-f006:**
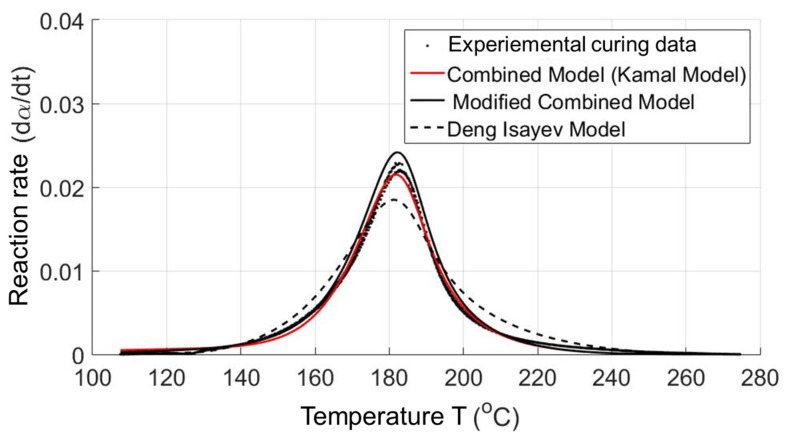
Modeling of cure kinetics models for the injection molding simulation.

**Figure 7 polymers-15-02984-f007:**
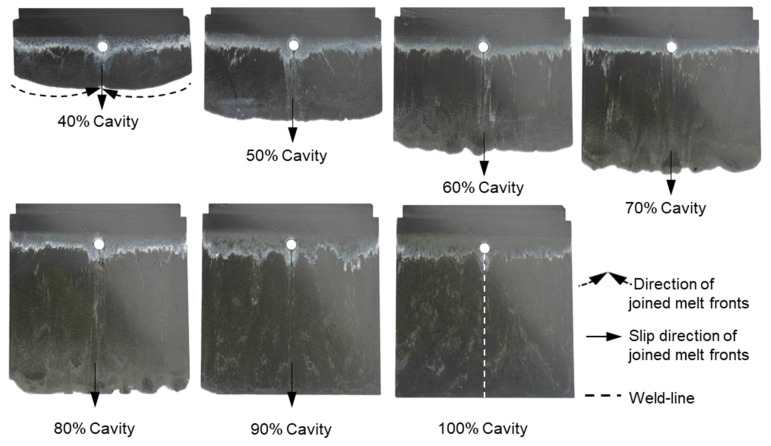
PF6680 (GF25 + GB30); the mechanism for reuniting two melt fronts behind an obstacle. Mold temperature is 175 °C, and injection speed is 8 cm^3^/s.

**Figure 8 polymers-15-02984-f008:**
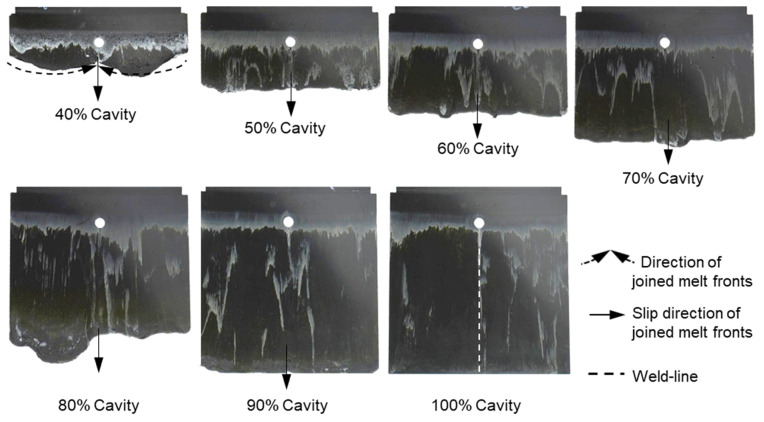
PF6506 (GF30 + GB30); the mechanism for reuniting two melt fronts behind an obstacle. Mold temperature is 175 °C, and injection speed is 8 cm^3^/s.

**Figure 9 polymers-15-02984-f009:**
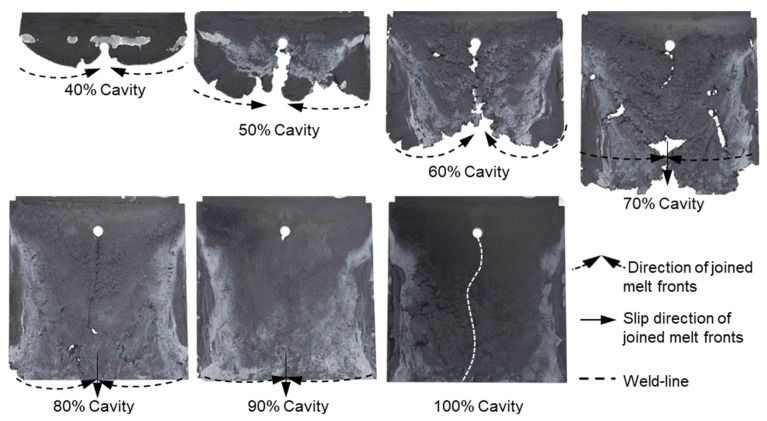
PF1110 (GF35 + GB45); the mechanism for reuniting two melt fronts behind an obstacle. Mold temperature is 175 °C, and injection speed is 8 cm^3^/s.

**Figure 10 polymers-15-02984-f010:**
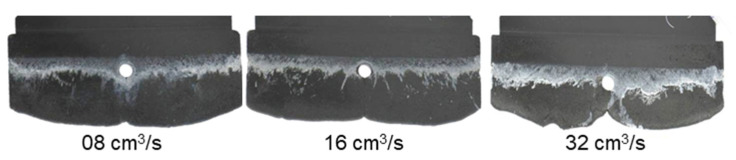
PF6680 (GF25 + GB30); the influence of injection speeds on the joining mechanism of the melt fronts behind an obstacle. Mold temperature of 175 °C is constant, and cavity volume is 40%.

**Figure 11 polymers-15-02984-f011:**
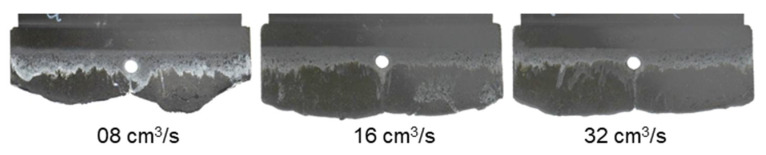
PF6506 (GF30 + GB30); the influence of injection speeds on the joining mechanism of the melt fronts behind the obstacle. Mold temperature of 175 °C is constant and Cavity volume is 40%.

**Figure 12 polymers-15-02984-f012:**
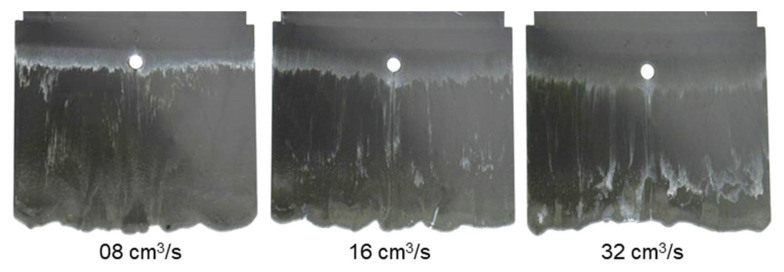
PF6680 (GF25 + GB30); the mechanism for reuniting melt fronts behind the obstacle. Mold temperature of 175 °C is constant, and cavity volume is 70%.

**Figure 13 polymers-15-02984-f013:**
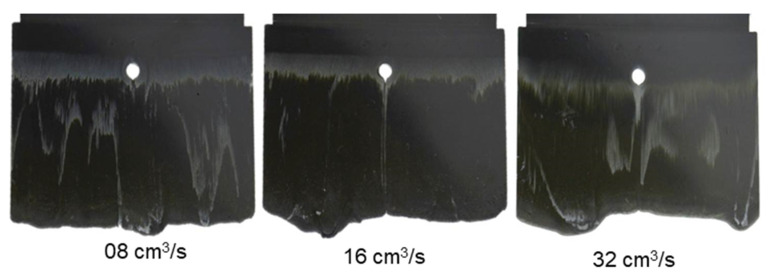
PF6506 (GF25 + GB30); the mechanism for reuniting melt fronts behind the obstacle. Mold temperature of 175 °C is constant, and cavity volume is 70%.

**Figure 14 polymers-15-02984-f014:**
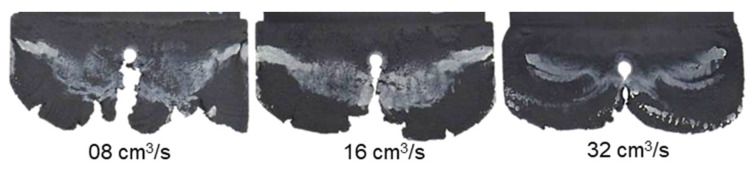
PF1110 (GF35 + GB45); the influence of injection speeds on the joining mechanism of the melt fronts behind an obstacle. Mold temperature of 175 °C is constant, and cavity volume is 50%.

**Figure 15 polymers-15-02984-f015:**
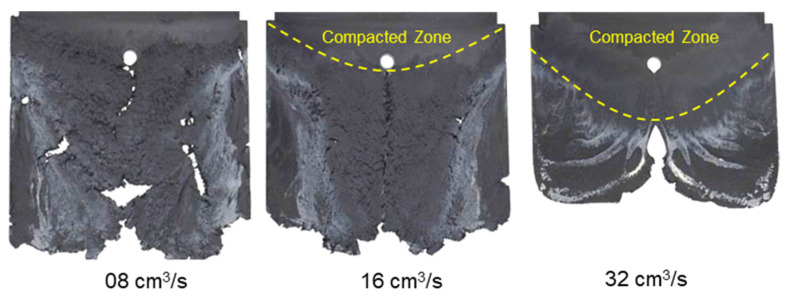
PF1110 (GF35 + GB45); the mechanism for reuniting melt fronts behind an obstacle. Mold temperature of 175 °C is constant, and cavity volume is 70%.

**Figure 16 polymers-15-02984-f016:**
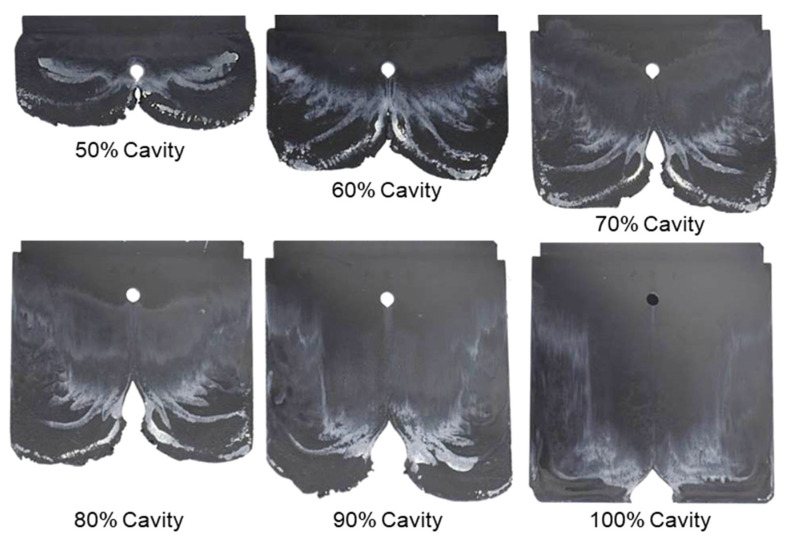
PF1110 (GF35 + GB45); the mechanism for reuniting two melt fronts behind an obstacle. Mold temperature is 175 °C, and injection speed is 32 cm^3^/s.

**Figure 17 polymers-15-02984-f017:**
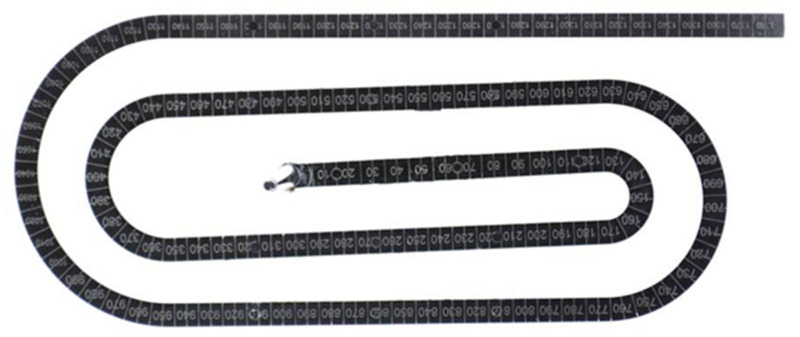
PF6680 (GF25 + GB30) and PF1110 (GF35 + GB45); the spiral molded part.

**Figure 18 polymers-15-02984-f018:**
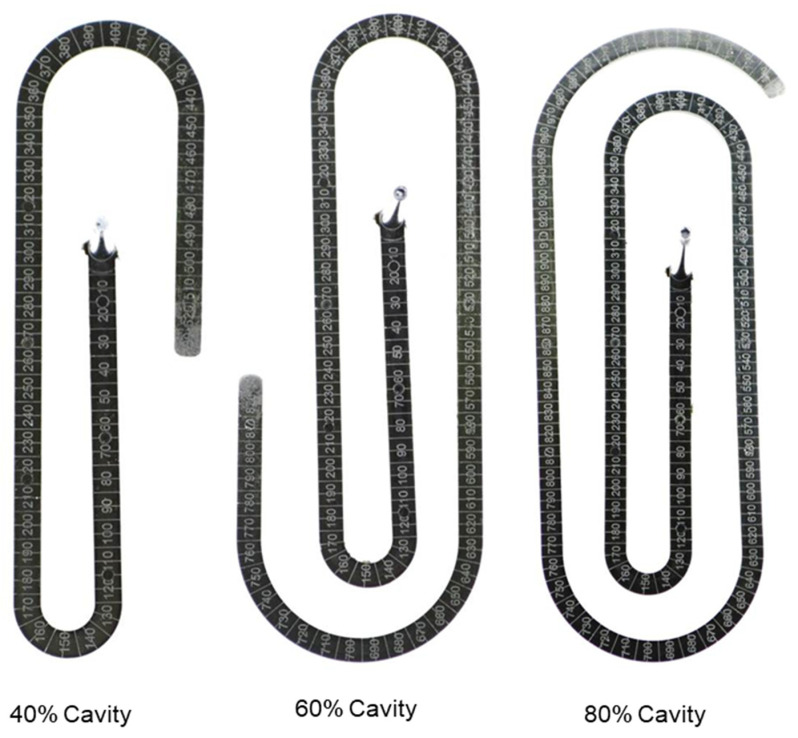
PF6680 (GF25 + GB30); the flow length of spiral molded part with different cavity volume.

**Figure 19 polymers-15-02984-f019:**
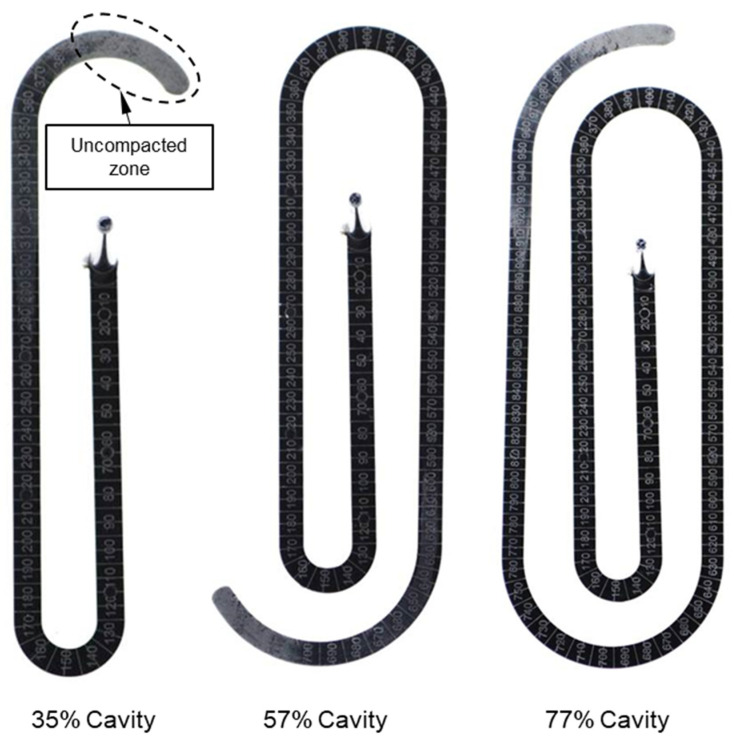
PF1110 (GF35 + GB45); the flow length of spiral molded part with different cavity volume.

**Figure 20 polymers-15-02984-f020:**
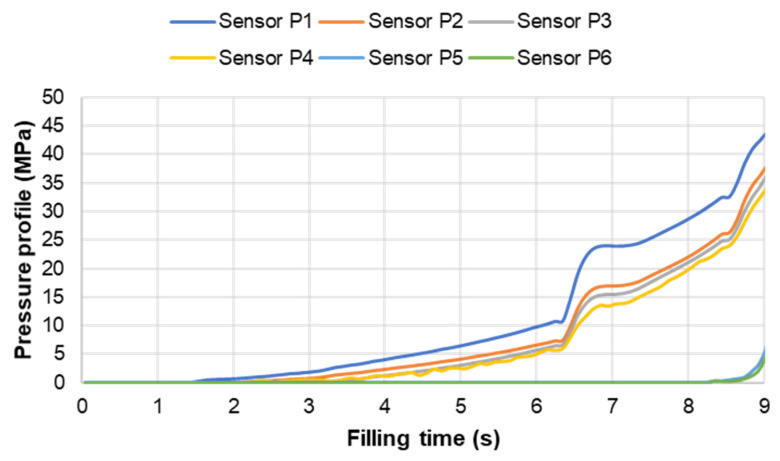
PF6680 (GF25 + GB30); cavity pressure profile in the filling phase.

**Figure 21 polymers-15-02984-f021:**
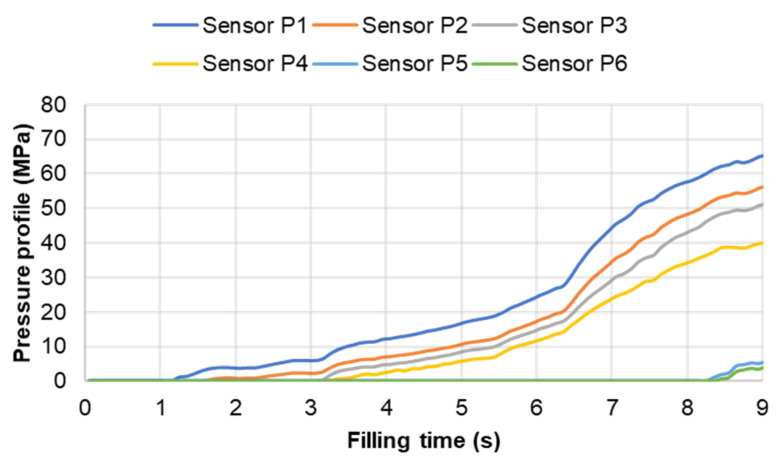
PF1110 (GF35 + GB45); cavity pressure profile in the filling phase.

**Figure 22 polymers-15-02984-f022:**
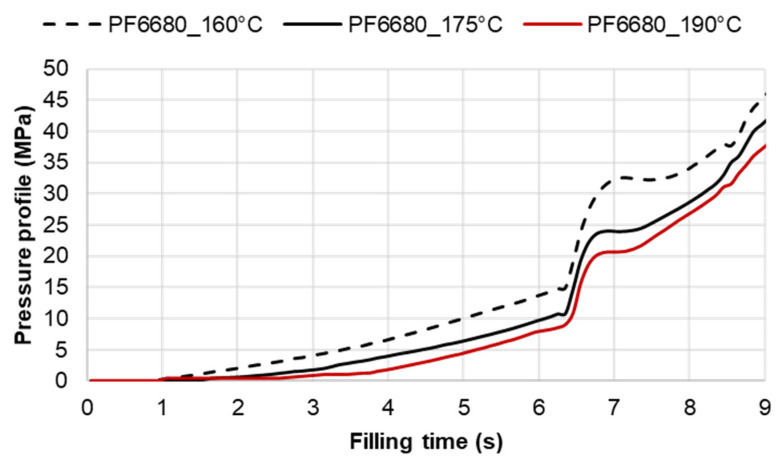
PF6680 (GF25 + GB35); pressure sensor 1; cavity pressure profile in the filling phase under different mold temperatures.

**Figure 23 polymers-15-02984-f023:**
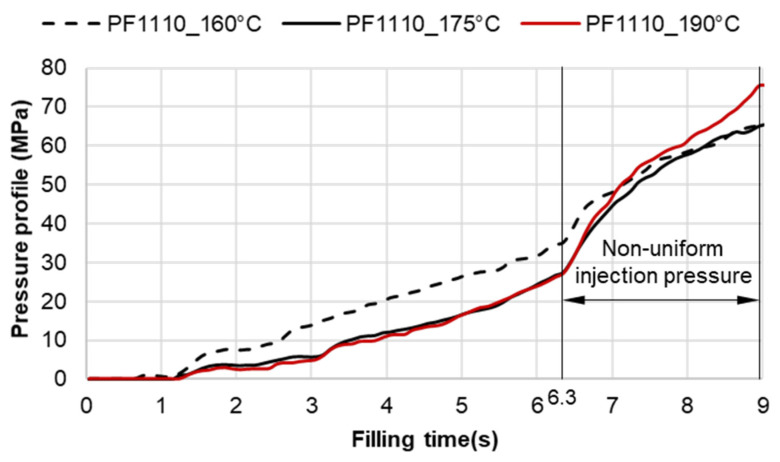
PF1110 (GF35 + GB45); pressure sensor 1; cavity pressure profile in the filling phase under different mold temperatures.

**Figure 24 polymers-15-02984-f024:**
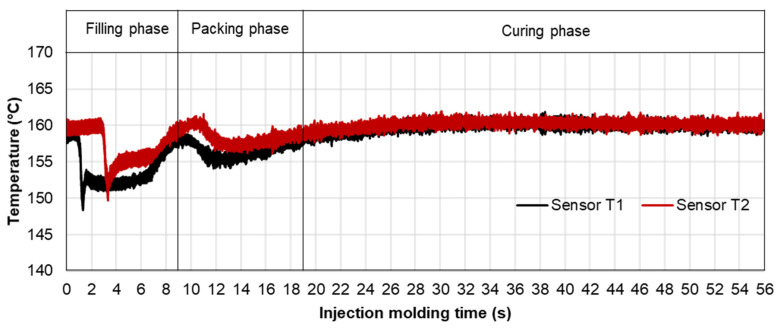
PF1110 (GF35 + GB45); infrared temperature sensor 1 (Sensor T1); infrared temperature sensor 2 (Sensor T2); the temperature distribution in the injection molding process.

**Figure 25 polymers-15-02984-f025:**
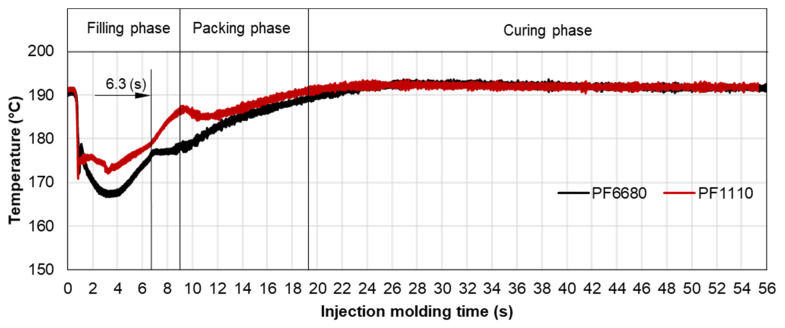
Infrared temperature sensor 1 (Sensor T1); comparison of the melt temperature between PF6680 and PF1110.

**Figure 26 polymers-15-02984-f026:**
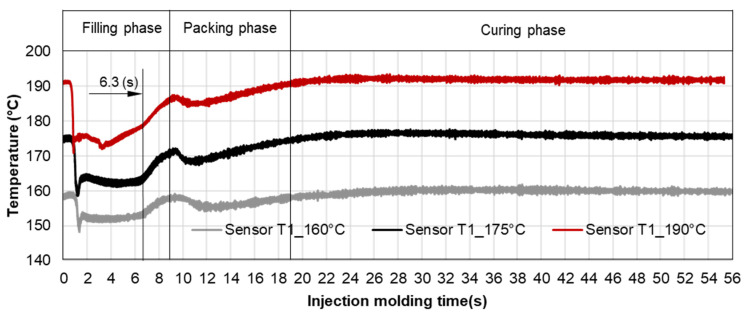
PF1110 (GF35 + GB45); infrared temperature sensor 1 (Sensor T1); the temperature distribution in the injection molding process under different mold temperatures.

**Figure 27 polymers-15-02984-f027:**
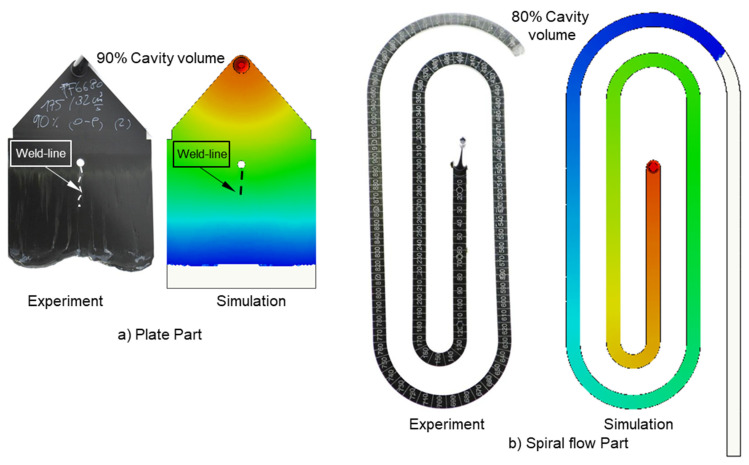
PF6680 (GF25 + GB30); comparison between simulation and experimental results.

**Figure 28 polymers-15-02984-f028:**
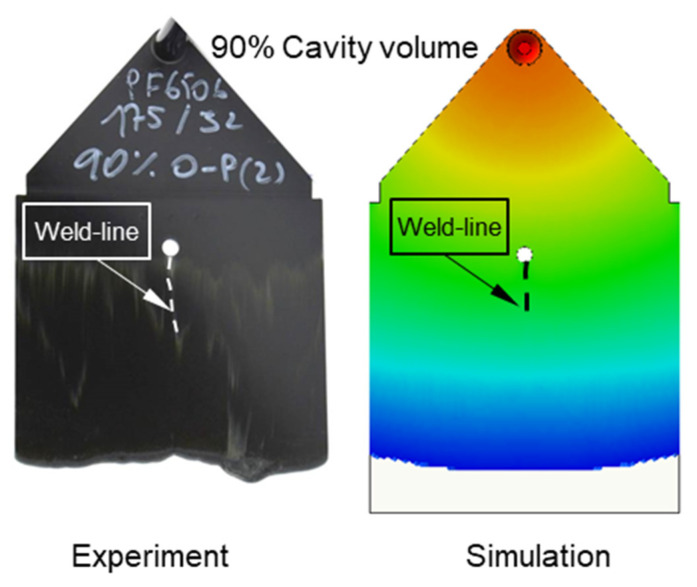
PF6506 (GF30 + GB30); comparison between simulation and experimental results.

**Figure 29 polymers-15-02984-f029:**
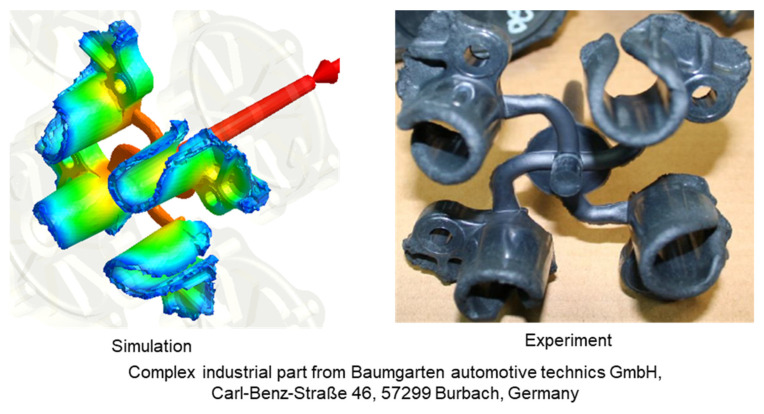
Complex industrial part; comparison between simulation and experimental results.

**Figure 30 polymers-15-02984-f030:**
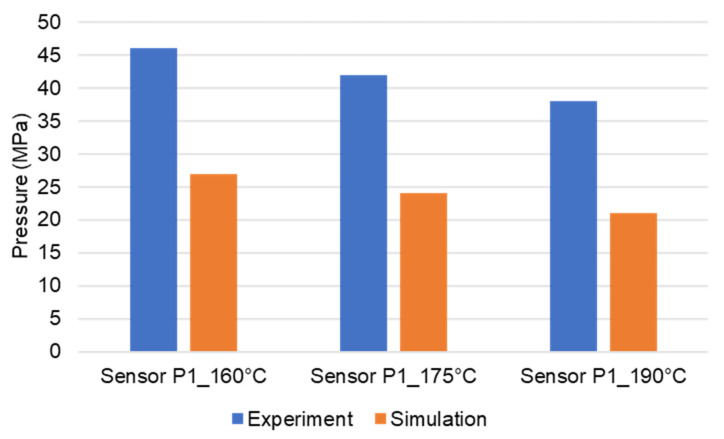
PF6680 (GF25 + GB30); spiral flow part. Comparison between injection pressure simulation and experimental results at the end of filling process.

**Figure 31 polymers-15-02984-f031:**
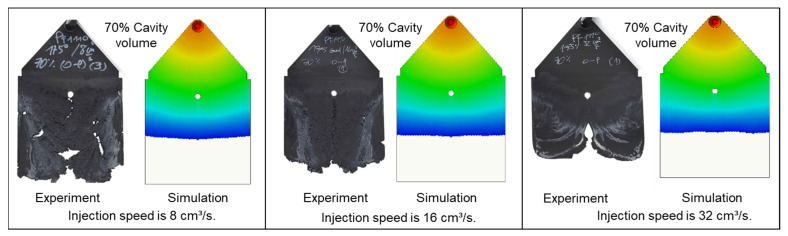
PF1110 (GF35 + GB45); comparison between simulation and experimental results.

**Figure 32 polymers-15-02984-f032:**
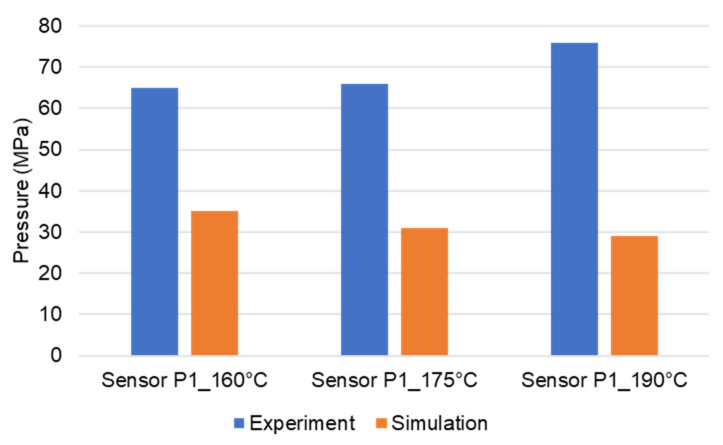
PF1110 (GF35 + GB45); spiral flow part. Comparison between injection pressure simulation and experimental results at the end of filling process.

**Table 1 polymers-15-02984-t001:** Experimental materials.

Abbreviation	Commercial Name	Manufacturer
PF-GF25 + GB30	Bakelite PF6680	Bakelite
PF-GF30 + GB30	Bakelite PF6506	Bakelite
PF-GF35 + GB45	Bakelite PF1110	Bakelite

## Data Availability

The data presented in this study are available on request from the corresponding author.

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
