# Peer review of "Flow Disturbance Characterization of Highly Filled Thermoset Injection Molding Compounds behind an Obstacle and in a Spiral Flow Part"

_polymers, 2023, doi:10.3390/polym15142984_

Round 1

Reviewer 1 Report

This manuscript studied the flow disturbance characterization of injection molded thermoset material on weldline and a spiral flow. According to your previous published paper, authors have done several researches on injection molded thermoset polymer which is seldom done. Good job on achieving the Global Innovation Talent Award 2018 at Moldex3d european meeting. However, my comments are as follows; 

1.      Line 77 surface roughness of what? Surface roughness of cavity and core?

2.      Line 113: Cp p is lower case.

3.      Line 169: viascosity

4.      Line 201: 55% to 80%?  Is it by volume or weight?

5.      Line 212: Where is the last part?

6.      Abbreviated experimental materials of GF25+GB30, Do they have any meaning of GF25 and GB30? Seems GF meaning glass fiber.

7.      What is the thickness of the gate closed to the sprue? It seems the thickness is uniform on Figure 3. For more uniform melt flow, the thickness near the sprue should be thicker than that near the film gate (Taper profile of thickness from sprue to the film gate). I suggest making a table to show your process parameters. In this case, it is easier for reader to read.

8.      Injection chamber is used to call as barrel on injection molding process.

9.      Can you show your mold heating channel layout? It seems you used oil as the heating medium. This is very important to make sure your mold temperature is uniform in turn melt flow is uniform too.

10.  Figures 20-23 Pressure gradient (Mpa) -> MPa, P must be capital, basic unit is Pascal. Pressure gradient (gradient means derivative) seems not a proper term, Maybe Cavity pressure profile is preferred.  

11.  The pressure gradient of Figures 23-25 was measured in where (P1~P6)? Although you have 6 pressure sensors.

Author Response

Dear Sir /Madam,

Thank you so much for your sincere support and giving me comments. After reading comments and your advice, a revised manuscript was corrected with additional information. Please find in the attached find my feedback.

Reviewer 2 Report

This research investigated the physical filling behavior of reinforced thermoset injection molding compounds, including the influence of wall slip and flow disturbance behavior on the mechanism of weld-line formation. Moreover, the optimal developed viscosity model was employed into commercial software to simulate the filling behavior of thermoset melt. Sufficient characterizations were performed and detailed analyses were also given in this manuscript. Only few doubts should be specified before accepting the publication of this manuscript in Polymers.

1. It is well known that the homogeneous dispersion and orientations of fillers in polymer matrix has big influence on the properties of polymer composites. Please clarify how about the filler’s distribution in phenolic matrix during filling process, especially for highly filled content.

2. For the thermoset injection molding compounds with the filler content of more than 65%, the authors gathered that the wall slip phenomenon has a great impact on melt filling behavior. Please explain the influence mechanism, and clarify how about the local melt viscosity variation due to the wall slip phenomenon?

3.  The figure 3 is not clear, please change a high-definition image.

4. please change the language of caption and legend of figure 5, figure 6 into English.

Minor editing of English language and few figure format are required.

Author Response

Dear Sir /Madam,

Thank you so much for your sincere support and giving me comments. After reading comments and your advice, a revised manuscript was corrected with additional information. Please find in the attached find my feedback.

If you have any questions, please do not hesitate to contact me.

Best regards

Dr.-Ing. Ngoc Tu Tran
